# Mechanisms of *Trichoderma longibrachiatum* T6 Fermentation against *Valsa mali* through Inhibiting Its Growth and Reproduction, Pathogenicity and Gene Expression

**DOI:** 10.3390/jof8020113

**Published:** 2022-01-25

**Authors:** Na Zhu, Jing-Jiang Zhou, Shu-Wu Zhang, Bing-Liang Xu

**Affiliations:** 1College of Plant Protection, Gansu Agricultural University, Lanzhou 730070, China; zhuna54@163.com (N.Z.); jjzhou@gsau.edu.cn (J.-J.Z.); 2Gansu Provincial Biocontrol Engineering Laboratory of Crop Diseases and Pests, Lanzhou 730070, China; 3State Key Laboratory of Green Pesticide and Agricultural Bioengineering, Ministry of Education, Guizhou University, Guiyang 550025, China; 4Gansu Provincial Key Laboratory of Arid Land Crop Science, Gansu Agricultural University, Lanzhou 730070, China

**Keywords:** *Trichoderma longibrachiatum* (T6), biological control, *Valsa mali*, pathogenicity

## Abstract

Apple Valsa canker is one of the most serious diseases, having caused significant apple yield and economic loss in China. However, there is still no effective biological methods for controlling this disease. Our present study focused on the inhibitory activity and mechanisms of *Trichoderma longibrachiatum* (T6) fermentation on *Valsa mali* that causes apple Valsa canker (AVC). Our results showed that the T6 fermentation exhibited effective antifungal activity on the mycelial growth and conidia germination of *V. mali*, causing mycelium malformation and the hyphal disintegrating in comparison to the control. The activity of pathogenically related enzymes that are secreted from *V. mali* and the expression level of gene of *V. mali* were significantly inhibited and downregulated by treatment with T6 fermentation. In addition, the lesion area and number of pycnidia of *V. mali* formed on the branches were significantly reduced after treatment with the T6 fermentation through the pathogenicity test on the detached branches. Our results indicate that the possible mechanism of T6 fermentation against *V. mali* occurs through inhibiting its growth and reproduction, the pathogenic enzyme activity, and its related gene expression.

## 1. Introduction

Apple Valsa canker is a devastating disease that caused by *Valsa mali* [1], which results in significant damages to apple production and huge economic losses in East Asia [2]. It is one of the important factors limiting apple production in Japan [3] and South Korea [4]. In China, apple Valsa canker is considered as one of the devastating diseases in apple orchards, restricting the yield and quality of apple fruits and causing great economic losses to the apple industry every year [5,6]. Although apple Valsa canker has been occurring and seriously damaging apple orchards, there is no effective control method [7].

Currently, chemical control is the main method for apple Valsa canker control [8,9,10], due to the limitation of resistant breeding against apple Valsa canker [11,12]. However, there are increasing concerns regarding to the impact of chemical fungicides on environmental pollution. Many chemical fungicides have been also challenged by the resistance development to pathogens [13]. Furthermore, *V. mali* mainly invades the phloem and xylem of apple trees, which makes it difficult to control with conventional fungicide treatments [14]. In order for a substantial increase in plants production to be maintained and for the growing demand for food and environmental quality to be met, biological agents have been widely used to control plant diseases because they are environmentally more friendly [15]. Some biocontrol agents used to control plant diseases have been reported widely in recent years [16,17]. *Trichoderma* is one of the widely distributed and broad-spectrum biocontrol fungi, i.e., the secondary metabolites of *Trichoderma* play an important role in controlling of plant disease [18,19]. A number of reports have revealed that the fermentation of *Trichoderma* had a significant inhibitory effect on different species of plant pathogens [20,21]. However, the previous studies mainly focused on the effect of the pathogens on mycelial morphology and growth, and only few reports have focused on the inhibitory effect of *Trichoderma longibrachiatum* T6 (T6) on *V. mali* gene expression and reproduction. T6 is a well-known biological control agent against several plant pathogens with limited harm to the environment. Our previous study reported that T6 fermentation has been shown to have an inhibitory effect on 11 pathogens (*Phaeoramularia capsicicol**a*, *Colletotrichum lindemuthianum*, *Fusarium semitectum*, *Rhizoctonia solani*, *Alternaria mali*, *Botrytis cinerea*, *Trichothecium roseum*, *Fusarium oxysporum*, *Aspergillus niger*, *Rhizopus nigricans*, *V. mali*) [22], and the average inhibitory rate of T6 strain on *Fusarium oxysporum* was 60.09% [23]. However, there is little information regarding to the mechanisms of T6 fermentation against *V. mali.*

A previous study reported that the mechanism for the infection of *V. mali* in the host was due to the secretion multiple of cell wall degrading enzymes (CWDEs) that include xylanase, β-glucosidase, cellulase (Cx), polygalacturonase (PG), and polymethylgalacturonase (PMG) [24]. Research has shown that the activity of polygalacturonase, polymethylgalacturonase, cellulase, β-glucosidase, and xylanase was closely related to *V. mali* pathogenicity during infection of apple tree [24,25]. In addition, some studies with cytochemistry, genomic, and transcriptome sequencing demonstrated that pectinase plays an important role in the infection process of *V. mali* [14,26,27]. The expression of the genes related to pectin catabolism, hydrolase activity, and biosynthesis of secondary metabolites increased during *V. mali* infection of apple tree [27]. The genes of *VmVeA, VmVelB Gvm2*, and *Gvm3* are involved in growth, conidial development, secondary metabolism, and virulence of *V. mali* [28,29]. However, there is little information regarding to the effect of T6 fermentation on the activities of CWDEs and its related gene expression at biochemical and molecular levels.

The purposes of this study were to determine the antifungal activity of T6 fermentation against *V. mali* at morphological, biochemical, and molecular levels; the effect of T6 fermentation on the pathogenicity, conidia germination, and mycelial growth of *V. mali*; the effect of T6 fermentation on the activity of CWDEs of *V. mali*; and the gene expression of *V. mali*. Our results will provide new insights into the biological control mechanism of T6 fermentation against *V. mali*.

## 2. Materials and Methods

### 2.1. Preparation of Pathogen and Antagonistic Strain 

*Valsa mali* and antagonistic strain of *Trichoderma longibrachiatum* (T6) were provided by the Laboratory of Plant Virology and Molecular Biology, Gansu Agricultural University. The antagonistic strain T6 is also deposited in the Institute of Microbiology, Chinese Academy of Sciences, with the accession number of CGMCC no.13183. The *V. mali* and T6 were grown on potato dextrose agar (PDA) (dextrose: China pharmaceutical group, Beijing, China; agar: Solarbio, Beijing, China) in Petri dishes for 5 and 6 days, respectively, at 25 °C with a 16 h light/8 h dark photoperiod regime.

### 2.2. Preparation of T6 Fermentation 

The conidia suspension of T6 was obtained by flushing the conidia of 6-day-old T6 from PDA culture with 5 mL of sterilized water into a sterile 10 mL centrifuge tube after adding a drop of Tween-80. The conidia were suspended by shaking and adjusted to 1.5 × 10^7^ conidia per milliter using a hemocytometer (Qiu Jing, Shanghai, China). The conidia suspension of T6 (1 mL) was mixed with 60 mL of sterilized liquid potato dextrose broth (PDB) culture medium in an Erlenmeyer flask (150 mL) and fermented for 7 days with shaking at 150 r/min and 28 °C. The fermentation was then filtered through a filter paper (Whatman Paper No. 3) and centrifuged at 12,000 r/min for 15 min at 4 °C. The supernatant was filtered with 0.22 μm Millipore membranes. Finally, the sterile T6 fermentation was stored as the stock solution at 4 °C for the experiments described below. 

### 2.3. Determination of Antifungal Activity of T6 Fermentation 

The sterile stock solution of T6 fermentation was diluted to 0-, 2-, 4-, 8-, 16-, and 32-fold with sterile water, and 0.5 mL of each diluted fermentation was added to 20 mL of PDA plate (8.5 cm in diameter). The same amount of sterile water in place of the fermentation was used as the control. The mycelial disc (5 mm in diameter) of 5-day-old *V. mali* culture was transferred to the center of each PDA plate and cultured at 25 °C. Six PDA plates were used for each fermentation dilution and sterile water. After being inoculated 5 days on the PDA medium at 25 °C, the growth of *V. mali* mycelium was observed and recorded every 24 h until *V. mali* was fully grew on the control plates without T6 fermentation. The level of the antifungal activity of T6 fermentation was calculated as the inhibitory rate (n) according to the formula n (%) = [(a − b)/a] × 100, where a is the colony area of *V. mali* without T6 fermentation on the control PDA medium, and b is the colony area of *V. mali* on the PDA medium treated with T6 fermentation [30]. The number of pycnidia produced on the colony of *V. mali* was recorded at 20 days of after inoculation. The experiment was performed with six replicates for each fermentation dilution and control. 

### 2.4. Examination of Conidia Germination and Mycelial Growth of V. mali

For the determination of conidia germination, the *V. mali* conidia were collected from the pycnidia on apple tree branches and suspended in the T6 fermentation at the concentration of 10^7^ conidia per milliliter conidia suspension. The conidia suspension (20 µL) was evenly placed on the PDA media with a sterilized applicator and then cultured at 25 °C for 0–30 h. The germination of the conidia was observed under a light microscope (Nikon, Japan). The germination rate was recorded when the length of germ tube was more than half of the diameter of conidia. Sterile water instead of the T6 fermentation was used as the control. 

To determine the mycelial growth of *V. mali* under different densities of T6 fermentation (dilutions at 0-, 2-, 4-, 8-, 16-, and 32-fold), we added 1 mL of each T6 fermentation dilution into a 150 mL Erlenmeyer flask containing 60 mL of PDB medium. A total of 1 mL of sterile water was used as control in a separate flask. Then, a 5-day-old *V. mali* mycelial disc (5 mm in diameter) was added into the PDB medium and inoculated with shaking at 25 °C and 150 r/min. The mycelium in either different T6 fermentation dilutions or water were obtained by filtration with sterile filter paper at 5 days after incubation. The mycelial morphology of *V. mali* was observed and photographed with light microscope (Nikon, Tokyo, Japan) and scanning electron microscope (Hitachi, Tokyo, Japan). The dry weight of the mycelium was recorded after air drying.

### 2.5. Determination of the Pathogenicity of V. mali on Detached Twig Branches

Two-year-old healthy apple branches of the “Fuji” variety were cut into 15 cm segments, washed twice with tap water, and disinfected with 75% alcohol. The twigs were then rinsed with sterile water 3 times, and both ends of the twigs were sealed with paraffin wax after they dried. The twigs were scalded with hot iron nail caps (5 mm in diameter) at a certain distance [31], each twig with three scalded areas. The 5-day-old mycelial discs (5 mm in diameter) of *V. mali* were inoculated on the scalded area, and the inoculated twigs were maintained in a moisture-controlled chamber at 25 °C and covered with fresh-keeping film. After 24 h, the T6 fermentation stock solution was sprayed on the inoculation sites before appearance of lesions. The same amount of sterile water was applied on the twigs as a control. The size of lesions and the number of pycnidia were measured and recorded regularly at 3, 5, 7, 9, 12, and 15 days after the inoculation. Three detached twigs were used for each treatment and control, separately, and the experiments were repeated three times.

### 2.6. Assay of the Cell Wall-Degrading Enzyme Activity of V. mali 

Three *V. mali* mycelial discs (5 mm in diameter) were taken from the edge of *V. mali* on PDA plates and added individually into each of the sterilized flasks (150 mL) containing 60 mL synthetic medium (SM). The SM contained KH_2_PO_4_ (0.4 g), MgSO_4_·7H_2_O (0.2 g), KCl (0.2 g), NH_4_NO_3_ (1 g), FeSO_4_ (0.01 g), ZnSO_4_ (0.01 g), and MnSO_4_ (0.01 g) in 1 L distilled water [32]. Thereafter, 1 mL of T6 fermentation stock solution was added in each flask, and the mixtures were incubated on the shaker at 150 r/min at 25 °C for 11 days, whereas 1 mL of sterile water was used as control. The culture filtrate (10 mL) was sampled from each flask at 3, 5, 7, 9 and 11 days by centrifuging at 8000 r/min at 4 °C for 15 min and was filtered through 0.22 μm Millipore membranes after incubation. The activities of pectinase, xylanase, and cellulase were determined with the dinitrosalicylic acid (DNS) method using 1% (*w/v*) pectin, xylan, and carboxymethyl cellulose as the substrates, respectively [33]. The culture filtrate (500 µL) was mixed with 2.5 mL of 1% (*w/v*) either pectin or xylan or carboxymethyl cellulose in 50 mM sodium citrate buffer (pH 5.0) and incubated at 50 °C for 30 min, and then 3 mL of the DNS solution was added to each reaction mixture and boiled for 5 min. The absorbance of the reaction mixtures was measured at 540 nm with a spectrophotometer. One unit of pectinase, xylanase, and cellulase activity (U) was defined as the amount of the enzyme that catalyses pectin, xylan, and carboxymethyl to 1 µmol of D-galacturonic acid, xylose, and glucose per minute, respectively. Each treatment and control were performed with six replications. 

### 2.7. Measurement of Gene Expression in V. mali before and after T6 Fermentation Treatment

Three mycelial discs were taken from a well-grown *V. mali* colony and put into the mixture of 60 mL of PDB medium and 1 mL of T6 fermentation and incubated at 25 °C and 150 r/min shaking. Sterile water was used instead of the T6 fermentation as the control. After cultivating for 5 days, the mycelium was collected by centrifuging at 12,000 r/min for 15 min and stored at −80 °C for later use.

Total RNA of the mycelium was extracted using the Fungal RNA Kit (Omega Bio-Tek, Norcross, GA, USA) and stored at −80 °C. RNA integrity was checked with 1% agarose gel electrophoresis and the radios of OD_260_/OD_280_ and OD_260_/OD_230_. The concentration of the total RNA was determined by UV spectrophotometer (Implen NanoPhotometer, Munich, Germany). The first-strand cDNA was synthesized according to the instructions of the first-strand cDNA synthesis kit (Takara Bio-Tek, Dalian, China) and stored at −20 °C. The expression level of 15 genes were detected by RT-qPCR. These genes were demonstrated by the previous transcriptome analysis that relate to the cell wall-degrading enzyme, growth, metabolism, and pathogenicity of *V. mali* [34]. The primers of the selected genes were designed by Primer Premier 5.0 and listed in Appendix A. The kits of TB Green^®^ Premix Ex Taq™ II (Takara Bio-Tek, Dalian, China) and instrument of applied biosystems quantstudio^TM^ 5 real-time PCR (Thermo Fisher Scientific, Waltham, MA, USA) were used for RT-qPCR reactions and gene expression determination, respectively. Total volume of reaction system was 20 μL, and the RT-qPCR analysis was repeated three times for each gene. The relative expression level was calculated by REST 2009 Software (Quiagen, Dusseldorf, Germany). The glucose-6-phosphate-dehydrogenase (*G6PDH*) and cyclophilin (*CYP*) genes were selected as the endogenous reference genes [35], and the Vm sample was used as control for the calculation of the relative expression levels.

## 3. Results

### 3.1. Inhibition of Colony Growth and Pycnidial Formation of V. mali by T6 Fermentation

The experimental results showed that T6 fermentation had a significantly inhibitory effect on the mycelial growth and pycnidia production of *V. mali* on PDA in Petri dishes (Figure 1, Table 1). Compared with the control, T6 fermentation significantly inhibited the *V. mali* growth after 5 days of inoculation. The colony diameter of *V. mali* treated with T6 fermentation was smaller than that of control under all treatments (Figure 1A). The highest inhibitory rates (94.86%) were observed on *V. mali* treated with the T6 fermentation without dilution. The inhibitory rates were 89.63% in *V. mali* treated with twofold dilution. The inhibitory rates decreased with the increase of dilution fold with the lowest inhibitory rate of 38.91% when the fermentation was diluted by 32-fold (Table 1). In addition, the number of pycnidia produced by *V. mali* treated with T6 fermentation was lower than that of the treated with sterile water after inoculation for 20 days (Figure 1B). The *V. mali* treated with stock solution of T6 fermentation (0-fold) did not produce any pycnidia on PDA at 20 days. With the increase of dilution fold, the number of pycnidia produced gradually increased. The numbers of pycnidia produced by *V. mali* were 22.67 ± 4.93 and 157.5 ± 6.63 after being treated with twofold and 32-fold dilution, respectively, while 227.83 ± 12.35 of pycnidia was produced by *V. mali* treated with sterile water (Table 1).

### 3.2. Inhibitory Effect of T6 Fermentation on the Mycelial Growth and Morphological Characters of V. mali 

T6 fermentation with different dilution had a significant inhibitory effect on the mycelial growth of *V. mali*. The measurements of mycelial dry weight showed that the mycelium grew slowest in the stock solution of fermentation with a mycelial dry weight of 56.53 ± 2.32 mg at 5 days, which was reduced by 75.73% in comparison to *V. mali* in the control group (sterile water). The dry weight increased with the increase of fermentation dilutions (Table 2). There was a significant difference in the dry weight between the control group (232.92 ± 8.53 mg) and the *V. mali* treated with 32-fold dilution fermentation (135.43 ± 6.68 mg) (*p* < 0.05). Microscopic observation revealed the T6 fermentation treatment also resulted in mycelium of *V. mali* was swelled and shrinkaged, and the protoplasm released (Figure 2C,F,G). It also enhanced hypha branching (Figure 2B), mycelium thickening, and mycelium cell wall rupturing and disintegrating (Figure 2D,H) compared with mycelium of *V. mali* culture alone (Figure 2A,E).

### 3.3. Inhibition of Conidia Germination of V. mali by T6 Fermentation

Both spores in control group and the spores inoculated with T6 fermentation for 6 h expanded into an ellipse but did not germinate (Figure 3(b1,b2)). After 12 h, spores began to germinate in sterile water (control group), being swollen but not germinated in fermentation treatment (Figure 3(c1,c2)). After 18 h, the spores in the T6 fermentation group began to germinate, but the germination rate (1.67%) with stock solution of T6 fermentation (0-fold) was lower than that of the control group (83.50%) (Figure 3(d1,d2)). It was 4.83% after the treatment with twofold dilution T6 fermentation. With the increase of dilution fold, the germination rates gradually increased to 50.17% when the fermentation was diluted by 32-fold (Table 2). The length of germ tube produced by T6-treated spores was shorter than that of the control after 24 h (Figure 3(e1,e2)). After 30 h of incubation, the hyphae in the control group were longer than those in the T6 fermentation-treated group (Figure 3(f1,f2)). The spores were treated with fermentation and sterile water for 0 h as control (Figure 3(a1,a2)).

### 3.4. Effect of T6 Fermentation on Lesion Development in Detached Twigs 

T6 fermentation had a significant inhibitory effect on the lesion development of *V. mali* on detached branches in comparison to the control, and there was a significant difference between them (Figure 4A,B). Compared with control, the area of the lesions on the treated branches was significantly smaller than that of control at 3, 5, 7, and 9 days after incubation (Figure 4A). The average area of the lesions in the treatment groups (treated with T6 fermentation) were decreased by 71.58%, 73.89%, 58.44%, and 54.56% at 3, 5, 7, and 9 days after incubation in comparison to the control group (treated with sterile water instead of T6 fermentation), respectively (Figure 4B).

Furthermore, T6 fermentation inhibited the production of pycnidia on the lesions after incubation from 9 to 15 days (Figure 4C). On the 9th and 12th days after inoculation, there were 28.33 ± 2.42 and 38.83 ± 3.19 pycnidia on the lesion surface in control group, respectively, while significantly less pycnidia (4.83 ± 0.75 and 6.50 ± 0.55) were produced on the lesion surface treated with T6 fermentation (*p <* 0.05). After 15 days of inoculation, the number of pycnidia in the treatment groups was 8.50 ± 1.05, which was 83.65% less than that of the control (Figure 4D). 

### 3.5. Cell Wall-Degrading Enzyme Activity of V. mali after T6 Fermentation Treatments

The activity of cell wall-degrading enzyme pectinase, cellulase, and xylanase of *V. mali* was significantly lower than that of the control during the cultivation process after treatment with T6 fermentation. The activity of pectinase was the highest, followed by cellulase, and then xylanase. The activity of pectinase decreased significantly in different periods after the fermentation treatment. The highest activity of pectinase reached 4.20 ± 0.33 U/mL on the 9th day, which was 49.28% lower than that of the control (Figure 5A). On the 11th day, the activity of pectinase was 3.02 ± 0.17 U/mL, which was significantly lower than control group (6.67 ± 0.05 U/mL) (*p* < 0.05). The activities of cellulase (Figure 5B) and xylanase (Figure 5C) were 0.91 ± 0.01 and 0.31 ± 0.03 U/mL, respectively, on fifth and seventh days in treated groups, which were 57.28% and 51.56% lower than those in the control, respectively. The activities of cellulase and xylanase of *V. mali* treated with T6 increased along with the culture time and reached the maximum value on the 11th day, being 2.45 ± 0.03 U/mL and 0.52 ± 0.08 U/mL, respectively. These were significantly lower than the control group (3.81 ± 0.16 U/mL and 0.88 ± 0.08 U/mL, respectively) (*p* < 0.05). 

### 3.6. Expression of Genes of V. mali after T6 Fermentation Treatments

The RT-qPCR results showed that T6 fermentation downregulated the expression of the genes involved in growth, conidiation, virulence, cell wall-degrading enzymes, and metabolic pathway of *V. mali* at 5 days after inoculation (Figure 6). Compared with control, the expression level of cell wall-degrading enzyme-related genes was downregulated by 0.70-fold, 0.54-fold, 0.21-fold, and 0.29-fold for xylanase I, Pectinase 2, β-glucosidase 1, and β-glucosidase 2, respectively; the growth, conidial development, secondary metabolism, and virulence related genes *VmVeA*, *VmVelB*, *Gvm2*, and *Gvm3* were downregulated by 0.35- to 0.63-fold; and the metabolism-related genes (*superoxide dismutase, catalase, citrate synthase, malate synthase*) and the pathogenic effectors *VmPxE1* were downregulated by 0.38- to 0.58-fold and 0.53-fold, respectively. 

## 4. Discussion 

This study focused on evaluating the antagonistic effect of *T. longibrachiatum* T6 (T6) fermentation against *V. mali.* Our results showed that the T6 fermentation could be used as an effective biocontrol agent to control the disease that caused by *V. mali* through inhibiting the growth of its colonies even reproduction and pathogenicity. Furthermore, our study demonstrated that T6 fermentation significantly decreased the activity of cell wall-degrading enzymes and downregulated the expression of genes related to cell wall-degrading enzyme, pathogenic factor, and growth metabolism of *V. mali* at physiological and molecular levels. 

*Trichoderma* has been applied as biocontrol agent against the plant pathogen, including *V. mali*, which has been confirmed in vitro [36,37]. Our results indicate that T6 fermentation presented great inhibitory potential against *V. mali*. In a laboratory test, different dilutions of T6 fermentation had a strong antifungal effect on the colony growth of *V. mali*. There were some other species of *Trichoderma* that exhibited antifungal effects on the colony growth and reproduction of other pathogens, i.e., *T. pseudokoningii*, which significantly inhibited the growth and sporulation of *B. cinerea* [38]; *T. harzianum* inhibited the growth and reproduction of some pathogenic fungi including *Fusarium* spp., *R. solani*, and *Macrophomina phaseolina* [39,40]. Interestingly, we found that T6 fermentation also decreased the number of pycnidia of *V. mali* significantly on PDA media. 

In addition, a number of studies have demonstrated that some biocontrol agents suppressed the plant pathogenic fungi by inhibiting their spore germination and mycelial growth. *Bacillus amyloliquefaciens* GB1 significantly reduced the conidia germination and the hyphal growth of *V. mali* by producing the antifungal metabolites [41]. In our present study, the conidia germination rates and mycelial growth were decreased by 98.1% and 75.7%, respectively, after treatment with T6 fermentation. In contrast, the antifungal activity of nonanoic acid against *Crinipellis perniciosa* and *Moniliophthora roreri* that was secreted from *T. harzianum* had 75% suppression of spore germination and 70% reduction of mycelial growth [42]. Furthermore, our results showed that T6 fermentation significantly decreased and delayed the spore germination, as well as suppressed the mycelial growth by rupturing and deforming the mycelial cell wall of *V. mali*. The possible mechanism may have been due to the production of cell wall-degrading enzymes and the antagonistic compounds from T6 fermentation. Previous studies reported that the cell wall-degrading enzyme of *Trichoderma* plays an important role in changing the morphological structure of mycelium and in inhibiting mycelial growth, including protease, chitinase, and glucanase [43,44]. Additionally, the secondary metabolites produced by *Trichoderma* including the diketopiperazines, ergosterol derivatives, and peptaibols had a significant antifungal effect on *B. cinerea* [38]. *Bacillus velezensis* was shown to have the ability to change the morphological structure of *V. mali* mycelium, including hyphal deformities, wrinkles, and ruptures [45]; our previous study found that the secondary metabolites (1,2-benzenedicarboxylicacid, bis-(2-methylpropyl)-ester (DIBP), (*Z*)-octadec-9-enoic acid, 1,2-benzenedicarboxylic acid, mono-(2-ethylhexyl)-ester (MEHP), and (*Z*)-13-docosenamide) of *T. longibrachiatum* T6 had inhibitory rates of 95% against *V. mali* [22]. Meanwhile, the volatile organic compounds (VOCs) produced by *Trichoderma* play a major role in inhibiting the mycelial growth of *Sclerotinia* spp. and *Fusarium oxysporum* [46].

Previous studies showed that smearing of *Saccharothrix yanglingensis* Hhs.015 (Hhs.015) fermentation on detached branches inhibited the lesion expansion and pycnidial production of *V. mali,* and the lesion expansion rate was reduced by 50% after treatment with Hhs.015 fermentation [47]. Similarly, the strain of *B. amyloliquefaciens* GB1 exhibited higher activity on decreasing of *V. mali* infection to apple branches [41]. In our study, T6 fermentation reduced the expansion area of lesions on detached branches and the number of pycnidia that was produced on the lesions significantly after application of T6 fermentation and inoculation of *V. mali* in comparison to the previous study. These findings provided a basis for the further field trials. The possible mechanism of T6 fermentation in controlling of apple Valsa canker disease on detached branches is response for the production of cell wall-degrading enzymes and antagonistic compounds, which decreased the pathogen (*V. mali*) growth, reproduction, and pathogenicity. A previous study showed that the gluconase, chitinase, and protease of *Aureobasidium pullulans* can suppress the growth and infection of *B. cinerea* and *Penicillium expansum* on apple fruit [48].

The mechanism of *V. mali* infection is complex, in which cell wall-degrading enzymes, secondary metabolites, and effector proteins play an important role in increasing its pathogenic activity [49,50,51], as well as promoting its invasion and colonization of host tissues [52]. In a preliminary study on the infection of *V. mali* on apple tree bark, it was found that the infection process was related to pectin catabolism, and that hydrolase activity and secondary metabolite biosynthesis, and four pectinases were highly abundant during infection [27,34,53]. In our present study, the results showed that T6 fermentation reduced the activity of cell wall-degrading enzymes (pectinase, cellulase, and xylanase) of *V. mali*. In particular, the enzyme activity of pectinase was significantly decreased by 49.28%, which indicates that T6 fermentation could reduce the pathogenicity, invasion, and colonization of *V. mali* on apple trees. It has been reported that *T. virens* ZT05 significantly inhibited the catalase and superoxide dismutase of *R. solani* [54]; Hhs.015 downregulated the glycoside hydrolase gene expression including pectinase genes of *V. mali* [34]. Our present study found that in the genes related to the cell wall-degrading enzymes of pectinase, cellulase, and xylanase, metabolism and pathogenicity of *V. mali* were significantly downregulated after inoculation with T6 fermentation.

## 5. Conclusions

T6 fermentation effectively exhibited inhibitory activity on *V. mali* growth, reproduction, and pathogenicity in vitro and detached branches in vivo. The possible mechanisms of T6 fermentation against *V. mali* are response for inhibiting the mycelial growth and spore germination, reducing the lesion expanding and pycnidia formation and decreasing the pathogenicity and pathogenic enzymes activity of *V. mali* through downregulating its related gene expression. More in-depth research is needed focus on developing an effective T6 agent in controlling of apple Valsa canker in the field.

## Figures and Tables

**Figure 1 jof-08-00113-f001:**
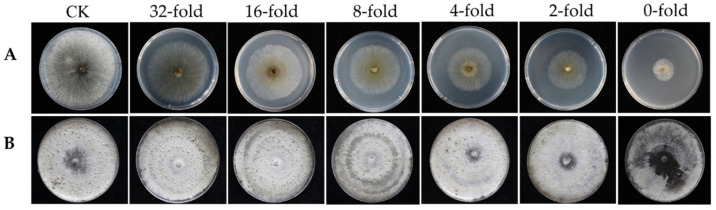
Inhibitory effect of different dilutions of *Trichoderma longibrachiatum* T6 fermentation on *Valsa mali* on PDA media. (**A**) Colony growth. (**B**) Pycnidia production. The dilution folds are indicated above images. CK is the control without T6 treatment.

**Figure 2 jof-08-00113-f002:**
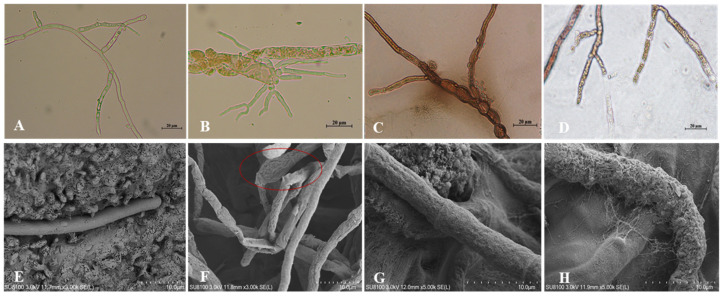
Micrographs of *Valsa mali* mycelium treated with *Trichoderma longibrachiatum* T6 fermentation stock solution for 5 days. Mycelium observed by light microscope (**A**–**D**) and scanning electron microscope (**E**–**H**). (**A**,**E**) Normal mycelium of *V. mali*. (**B**) Hypha to branch. (**F**) Treated mycelium with swelling (marked in red circle), shrinkage. (**C**,**G**) Protoplasm released. (**D**,**H**) Treated mycelium with hyphal disintegration.

**Figure 3 jof-08-00113-f003:**
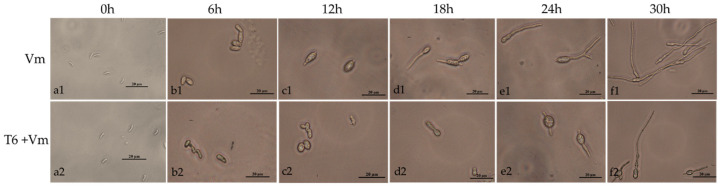
Inhibitory effect of *Trichoderma longibrachiatum* T6 fermentation on spore germination of *Valsa mali* in different time periods. The top panel (Vm) represents the spores germinated in sterile water (**a1**–**f1**). The bottom panel (T6 + Vm) represents the spores germinated in T6 fermentation (**a2**–**f2**). The treatment time was indicated above images.

**Figure 4 jof-08-00113-f004:**
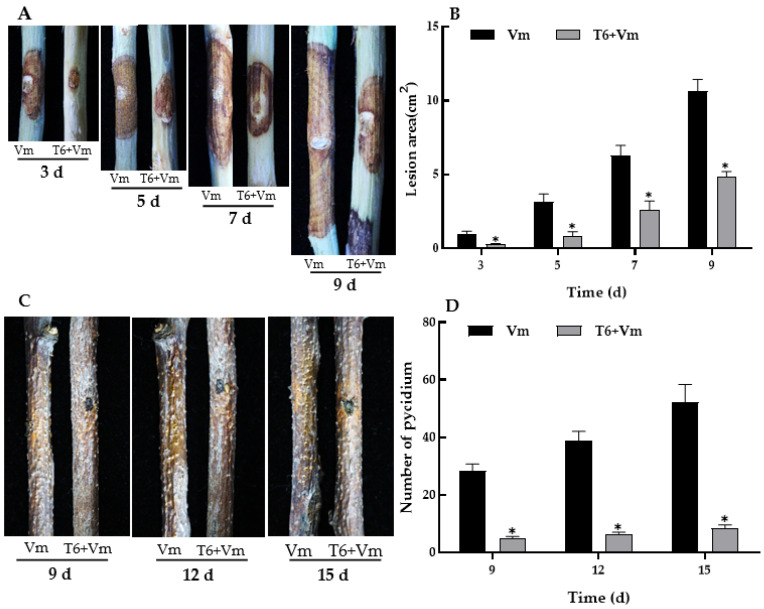
Effect of *Trichoderma longibrachiatum* T6 fermentation on pathogenicity of *Valsa mali*. (**A**) Development of lesion on the detached branches infected with *V. mali* treated with T6. (**B**) Quantification of the lesion area. (**C**) Formation of pycnidia on the detached branches infected with *V. mali* treated with T6 fermentation. (**D**) The number of pycnidia formed. Small bars represent the standard deviations of the means. Each test was repeated three times. Asterisk indicates the significant difference between the control group and the treatment group (*p <* 0.05).

**Figure 5 jof-08-00113-f005:**
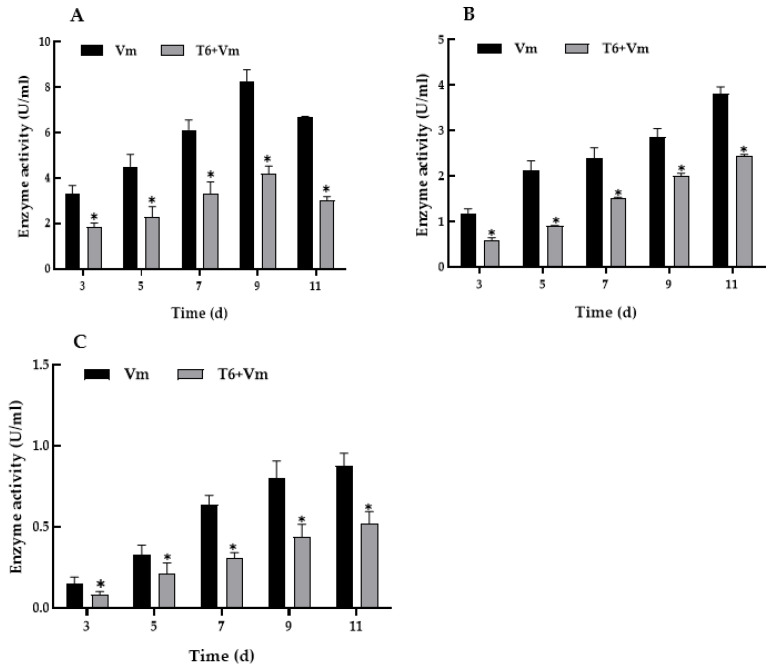
The activity of cell wall-degrading enzyme of *Valsa mali* treated with *Trichoderma longibrachiatum* T6 fermentation at stock solution. (**A**) Pectinase activity. (**B**) Cellulase activity. (**C**) Xylanase activity. The “T6 + Vm” represents liquid medium inoculated with *Valsa mali* and T6 fermentation. “Vm” represents liquid medium inoculated with *Valsa mali* only. The abscissa indicates that the enzyme activity of *V. mali* was measured at 3, 5, 7, 9, and 11 days after T6 fermentation treatment. Small bars represent the standard deviations of the means, and an asterisk indicates the significant difference between the control group and the treatment group (*p <* 0.05).

**Figure 6 jof-08-00113-f006:**
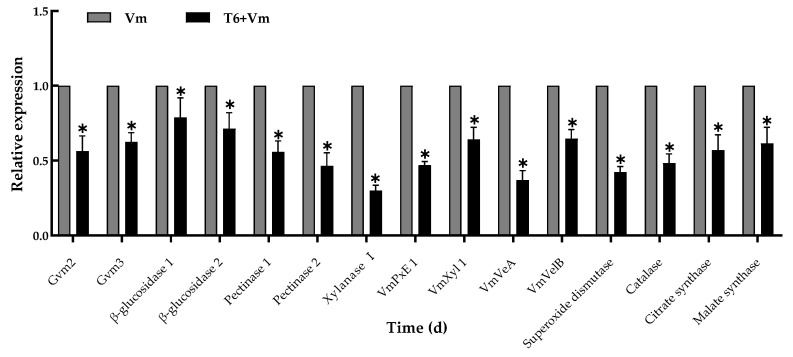
Relative expression level of 15 genes of *Valsa mali* after treatment with *Trichoderma longibrachiatum* T6 fermentation at stock solution for 5 days. The relative expressions represented in the histogram are the average values of three replicates. Asterisks indicate the significant difference between the control group and the treatment group (*p <* 0.05), and the small bars represent the standard deviations. The expression level of the control samples (Vm) without T6 treatment was normalized to 1.

**Table 1 jof-08-00113-t001:** Effect of different dilutions of *Trichoderma longibrachiatum* T6 fermentation on the colony and pycnidia production of *Valsa mali* on the PDA media after 20 days incubation.

Dilutions (Folds)	Colony Diameters (cm)	Inhibitory Rates (%)	Pycnidia Amounts (Pycnidia per Media)
0	1.93	94.86 ± 0.47 ^a^	0.00 ± 0.00 ^g^
2	2.73	89.63 ± 1.19 ^b^	22.67 ± 4.93 ^f^
4	4.17	75.97 ± 0.70 ^c^	41.17 ± 4.02 ^e^
8	4.67	69.85 ± 0.98 ^d^	59.50 ± 6.09 ^d^
16	5.61	56.46 ± 1.42 ^e^	125.00 ± 9.38 ^c^
32	6.64	38.91 ± 3.26 ^f^	157.50 ± 6.63 ^b^
ck	8.50	—	227.83 ± 12.35 ^a^

Value are means ± standard deviations of six replicates, and the different lowercase letters in the same column are significantly different at *p <* 0.05 according to Duncan’s new multiple range test and least significant difference test. The inhibitory rates (%) and pycnidia amount were determined on the 5th and 20th days after inoculation with *V. mali*. “0” represents stock solution of T6 fermentation. ck represents sterile water instead of T6 fermentation.

**Table 2 jof-08-00113-t002:** The inhibitory effect of *Trichoderma longibrachiatum* T6 fermentation against mycelial growth and conidia germination of *Valsa mali*.

Dilution (Folds)	Germination Rates (%)	Mycelial Dry Weight (mg)
0	1.67 ± 0.52 ^g^	56.53 ± 2.32 ^g^
2	4.83 ± 1.17 ^f^	77.33 ± 2.06 ^f^
4	8.83 ± 0.75 ^e^	94.40 ± 2.20 ^e^
8	15.17 ± 1.94 ^d^	105.18 ± 3.01 ^d^
16	33.17 ± 3.43 ^c^	115.63 ± 2.65 ^c^
32	50.17 ± 4.26 ^b^	135.43 ± 6.68 ^b^
ck	83.50 ± 3.21 ^a^	232.92 ± 8.53 ^a^

The value are means ± standard deviations of six replicates and the different lowercase letters in the same column are significantly different at *p <* 0.05 according to Duncan’s new multiple range test and least significant difference test. The mycelial dry weight was determined at 5 days after treated with T6 fermentation, and the germination rates (%) were determined at 18 h after being treated with T6 fermentation. ck represents sterile water instead of T6 fermentation. “0” represents stock solution of T6 fermentation.

## Data Availability

The data in this study are available on request from the corresponding author/first author.

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
