# Peer review of "Mechanisms of Trichoderma longibrachiatum T6 Fermentation against Valsa mali through Inhibiting Its Growth and Reproduction, Pathogenicity and Gene Expression"

_jof, 2022, doi:10.3390/jof8020113_

Round 1
Reviewer 1 Report
The objectives of this study were to determine the effect of T6 fermentation of Trichoderma longibrachiatum on pathogenicity, conidia germination and mycelial growth of Valsa mali, as well as the activity of CWDEs and the expression level of CWDEs encoding genes in V. mali. These studies are expected to provide new insight into the biological control mechanism of T6 fermentation against the Valsa apple cancer. These studies are not completely innovative, but important from an application point of view as they can improve the management of apple orchard disease caused by V. mali. Despite the importance of the topic, there are many shortcomings that should be addressed before publication.
The main comments are provided below, and the detailed comments are provided on the PDF version of the manuscript:
- the Authors should supplement the research with the analysis of the expression of selected genes at the same, several time points in which the enzyme activities were tested;unfortunately, the Authors only studied gene expression at one time point - please explain;
- please explain on what basis the reference gene was selected;
- the quality of photos and figures should be improved;
- the legend under the photos and figures should be detailed
;
- English style and grammar should be verifiedespecially in the "Discussion" section;
- the discussion should be rewritten;many errors, overly obvious and overlapping phrases were found;
- conclusions sound rather like a summary of the results and do not support the actual results, are too exaggerated - the comment mainly concerns the statement: "The results of this study provide new ideas for the development of new potential biocontrol agents against on V. mali".

Author Response
Responses to Reviewer 1
December 26, 2021
Dear Editors and Reviewer,
We would like to thank you for your comments/suggestions on the manuscript "Mechanisms of Trichoderma longibrachiatum T6 fermentation against Valsa mali through inhibiting its growth and reproduction, pathogenicity and genes expression". We have carefully revised by following your comments and suggestions, one-by-one. We believe that all the concerns have been adequately addressed in the revision. We hope the revised manuscript is now acceptable for publication in "Journal of Fungi".
Below are the details of our revision:
1.Please provide references.
Reply: Thanks for your valuable comments. We have added the relevant references in Line 59.
- The Authors should supplement the research with the analysis of the expression of selected genes at the same, several time points in which the enzyme activities were tested; unfortunately, the Authors only studied gene expression at one time point - please explain; It is recommended to use two reference genes. Is the stability of the selected gene appropriate in the tested experimental setup? Has such a validation been carried out?
please explain on what basis the reference gene was selected;
Reply: Thank you for your valuable suggestion. The genes of cell wall degrading enzyme, growth, metabolism, pathogenicity of V. mali were selected to conduct the expression. We have found that the optimal time for the effect of T6 fermentation on the colony, mycelium morphology, mycelium growth, pathogenicity and activity of cellulase and xylanase (5th and 7th day) of V. mali on the 5th day according to the experimental results. Therefore, we have considered the 5th day was the optimal time for genes expression analysis. In order to obtain the significant expression genes (cell wall degrading enzyme, growth, metabolism, pathogenicity of V. mali), and analyze or compare their expression at the same time point, the expression level of genes of cell wall degrading enzyme, growth, metabolism, pathogenicity of V. mali was determined at the 5 days.
The reference gene was selected in present study according to the published papers by Yin et al. (2013). The G6PDH gene was the most appropriate and stable reference gene in the previous and our present experiments. Yes, the stability of the selected gene appropriated in the tested experimental setup in each repeat. Therefore, we have used one reference gene to carry out the RT-qPCR analysis.
4.Higher resolution and quality of photos would be recommended. The legend under the photos and figures should be detailed.
Reply: Thank you for your valuable suggestions. We have provided higher resolution and quality of in the revised manuscript. The legend under the photos and figures were detailed one-by-one (line 220, 257, 279, 310).
5.Please explain what the values 3, 5, 7, 9, 11 mean.
Reply: Thank you. We have explained in the revised manuscript. The values 3, 5, 7, 9, 11 indicate that the enzyme activity of V. mali was measured at 3, 5, 7, 9, and 11 days after treatment with T6 fermentation (lines 346 to 351).
6.In my opinion, the post-culture fluid of the T6 strain. (It is mentioned in the discussion and conclusion)
Reply: Thank you for your valuable suggestions. We have changed “T6 strain” to “T6 fermentation” in the revised manuscript (line 377 and line 459).
7.Some statements are rather too obvious.
Reply: Thank you for your valuable suggestions. We have rewritten the statements. (line 386 to 405).
8.This work (Zhang et al. 2018) confirms that the current study is not innovative.
Reply: Thank you for your comments. In the previous study (Zhang et al. 2018), the authors determined the inhibitory effect of the metabolites produced by Trichoderma longibrachiatum T6 on colony growth of V. mali only, but they never determined the effect of T6 fermentation on the mycelium growth, reproduction, pathogenicity of V. mali, and its related genes expression. Thus, our present work is an in-depth study for the previous work, and we discovered another possible mechanism of T6 fermentation against V. mali through inhibiting its growth and reproduction, the pathogenic enzymes activity and its related genes expression.
9.Please clarify that. Morphology is morphology whatever it may be. It cannot be destroyed, only distorted / changed in relation to the "typical" one.
Reply: Thank you for your valuable suggestion. We have corrected the mistake in the revised manuscript (line 408).
10.The discussion should be rewritten; many errors, overly obvious and overlapping phrases were found.
Reply: Thank you for your suggestions. We have rewritten the section of discussion and supplement some references in the revised manuscript. In addition, the rewritten section has improved and edited by the corresponding author and a visiting professor from a visiting professor from the Rothamsted Research, Harpenden, UK.
11.Conclusions sound rather like a summary of the results and do not support the actual results, are too exaggerated - the comment mainly concerns the statement: "The results of this study provide new ideas for the development of new potential biocontrol agents against on V. mali".
Reply: Thank you for your suggestions. We have rewritten this section according to your wonderful suggestion (lines 459 to 466).
In addition to your suggestions and comments, a few other minor changes have been made to make the paper read better. Thank you!

Reviewer 2 Report
Some Minor points need to be addressed as follow:
Line 65, 70, 352: use citation of JoF format
Line 123, 134, 135, 177: Use JoF format (company, state, country)
Line 213, 234, 242, 260 check spelling of Trichoderma longibrachiatum
Line 285: shift "D" into the left panel of graph
Line 365: Some?
Line 360-362: Discuss more detail why T6 fermentation delay spore germination
Line 368-369: For instance? What kind of secondary metabolites? How about volatile compounds?
Line 372: Saccharothirx yangliensis remove "."
Line 375-376: Explain why?
Line 393-394: check font size
Line 400-402: T6 fermentation down-regulated CWDEs by itself or secondary metabolites in T6 fermentation down-regulated? please indicate more clearly

Author Response
Responses to Reviewer 2
December 26, 2021
Dear Editors and Reviewer,
We would like to thank you for your comments/suggestions on the manuscript "Mechanisms of Trichoderma longibrachiatum T6 fermentation against Valsa mali through inhibiting its growth and reproduction, pathogenicity and genes expression". We have carefully revised by following your comments and suggestions, one-by-one. We believe that all the concerns have been adequately addressed in the revision. We hope the revised manuscript is now acceptable for publication in "Journal of Fungi".
Below are the details of our revision:
1.Line 65, 70, 352: use citation of JoF format
Reply: Thanks for your valuable comments. We have revised it in the revised manuscript (line 65, 70, 413).
2.Line 123, 134, 135, 177: Use JoF format (company, state, country)
Reply: Thanks for your valuable comments. We have revised it in the revised manuscript (line 124, 135, 178).
Line 213, 234, 242, 260 check spelling of Trichoderma longibrachiatum.
Reply: Thanks for your valuable comments. We have revised it in the revised manuscript (line 220, 244, 257, 280).
Line 285: shift "D" into the left panel of graph.
Reply: Thanks for your valuable comments. We have revised it in the revised manuscript (line 306).
Line 365: Some?
Reply: Thanks for your valuable comments. We have changed “some” to “previous study“.
Line 360-362: Discuss more detail why T6 fermentation delay spore germination
Reply: Thanks for your valuable comments. We have explained and detailed the reason of T6 fermentation delay spore germination in the revised manuscript (lines 403 to 418). The possible mechanism may due to the production of cell wall degrading enzymes and the antagonistic compounds from T6 fermentation. Previous studies reported that cell wall degrading enzymes of Trichoderma play an important role in changing the morphological structure of mycelium and inhibiting mycelium growth, such as protease, chitinase and glucanase [45,46]. Additionally, the secondary metabolites produced by Trichoderma had a significant inhibitory effect on plant pathogenic fungi, such as diketopiperazines, ergosterol derivatives, peptaibols, etc [40]. Bacillus velezensis inhibited and changed the morphological structure of V. mali mycelium including hyphal deformities, wrinkles, and ruptures [47]; our previous study found that the secondary metabolites (1,2-Benzenedicarboxylicacid, bis-(2-methylpropyl)-ester (DIBP), (Z)-octadec-9-enoic acid, 1,2-Benzenedicarboxylic acid, mono-(2-ethylhexyl)-ester (MEHP) and (Z)-13-Docosenamide) of T. longibrachiatum T6 had broad spectrum and patent activity against pathogens with the inhibition rate of 95% against V. mali [22].
Line 368-369: For instance? What kind of secondary metabolites? How about volatile compounds?
Reply: Thanks for your valuable comments. We have supplied the kind of secondary metabolites that produced from Trichoderma in the revised manuscript (lines 412 to 420).
The secondary metabolites produced by Trichoderma had a significant inhibitory effect on plant pathogenic fungi, such as diketopiperazines, ergosterol derivatives, peptaibols, etc [40]. Our previous study found that the secondary metabolites (1,2-Benzenedicarboxylicacid, bis-(2-methylpropyl)-ester (DIBP), (Z)-octadec-9-enoic acid, 1,2-Benzenedicarboxylic acid, mono-(2-ethylhexyl)-ester (MEHP) and (Z)-13-Docosenamide) of T. longibrachiatum T6 had broad spectrum and patent activity against pathogens [22]. Additionally, the volatile organic compounds (VOCs) produced by Trichoderma play a major role in antagonism of pathogenic fungi. GC-MS analysis of the pure cultures of one of the endophytic fungi Trichoderma longibrachiatum and the plant pathogens revealed the presence of several VOCs including hydrocarbons, alcohols, ketones, aldehydes, esters, acids, ethers and different classes of terpenes [47] (Rajani et al. Inhibition of plant pathogenic fungi by endophytic Trichoderma spp. through mycoparasitism and volatile organic compounds. 2020)
- Line 372: Saccharothirx yangliensis remove "."
Reply: Thanks for your valuable comments. We have removed it in the revised manuscript (line 421).
Line 375-376: Explain why? (Our results showed that T6 fermentation can reduce the expansion area of lesions on detached branches and the number of pycnidia that produced on the lesions.)
Reply: Thanks for your valuable comments. We have explained it in the revised manuscript (line 425 to 431). The mechanism may due to the production of cell wall degrading enzymes and the antagonistic compounds decreased the pathogen (V. mali) growth, reproduction and pathogenicity after application of T6 fermentation. Previous study showed that the lytic enzymes of gluconase, chitinase, and protease that can inhibit the growth and infection of pathogens [49], so as to inhibit the infection and expansion of pathogens on bark.
- Line 393-394: check font size
Reply: Thanks for your valuable comments. We have revised it in the revised manuscript (line 439).
Line 400-402: T6 fermentation down-regulated CWDEs by itself or secondary metabolites in T6 fermentation down-regulated? please indicate more clearly.
Reply: Thanks for your valuable comments. It is T6 fermentation. We have revised in the revised manuscript (line 441).
In addition to your suggestions and comments, a few other minor changes have been made to make the paper read better. Thank you!

Round 2
Reviewer 1 Report
Dear Authors, thank you for the corrections made. Thank you for your explanations regarding gene expression analysis and time point and reference gene selection. Nevertheless, I believe that two reference genes should be used and the results of their stability documented in the manuscript. The current version of the manuscript requires a few more corrections - methodology, discussion, conclusions (marked in the PDF file). I recommend linguistic proofreading using a specialized English language service. All in all, the current version needs some minor improvements.

Author Response
Responses to Reviewer 1
January 11, 2022
Dear Editors and Reviewer,
We would like to thank you for your comments/suggestions on the manuscript "Mechanisms of Trichoderma longibrachiatum T6 fermentation against Valsa mali through inhibiting its growth and reproduction, pathogenicity and genes expression". We have carefully revised by following your comments and suggestions, one-by-one. We believe that all the concerns have been adequately addressed in the revision. We hope the revised manuscript is now acceptable for publication in "Journal of Fungi".
Below are the details of our revision:
- I believe that two reference genes should be used and the results of their stability documented in the manuscript. The current version of the manuscript requires a few more corrections - methodology, discussion, conclusions (marked in the PDF file). I recommend linguistic proofreading using a specialized English language service. All in all, the current version needs some minor improvements.
Reply: Thank you for your valuable suggestions. We have supplemented the two reference genes (G6PDH and CYP) and re-analyzed the gene expression, and the gene expression level has been calculated by software REST in the experiment (Line 364). And some errors in the methods, discussions and conclusions marked in the PDF and errors in English grammar have been modified according to the comments. As for English language proofreading, it has improved and edited again by the corresponding author and a visiting professor from a visiting professor from the Rothamsted Research, Harpenden, UK.
- Please complete the details - manufacturer's name, country, city.
Reply: Thank you for your valuable suggestions. The manufacturer's name, country, city have been completed in the re-revised manuscript (Line 88, 89, 95)
- Please verify whether it is 5- (as above) or 6- day old cultures.
Reply: Thank you for your valuable suggestions. It is 6- day old cultures. Because more spores were produced on 6th day (Line 92).
- Please complete the details - name of the instrument, software, manufacturer's name, country, city.
Reply: Thank you for your valuable suggestions. The name of the instrument, software, and manufacturer's name, country, city have been completed in the manuscript (Line 186 to line 191).
- “in vitro”italic
Reply: Thank you for your valuable suggestions. The “in vitro” have been italicized (Line 386).
- Please specify what this potential is: " inhibitory potential"?
Reply: Thank you for your valuable suggestions. It is inhibitory potential (Line 387).
- Word "inhibition/inhibited etc." used too often. Please replace with synonyms.
Reply: Thank you for your valuable suggestions. The word "inhibition/inhibited etc." has been replaced with synonyms (Line 398, 401, 403, 405).
- Rewrite “patent activity against pathogens”
Reply: Thank you for your valuable suggestions. It has been revised. (Line 418)
- Please correct your English grammar.
Reply: Thank you for your valuable suggestions. English grammar has been corrected. (Line 429-434, line 445-451, line 457-463).
